# Towards Learning Convolutions from Scratch

Behnam Neyshabur
Google
neyshabur@google.com

## Abstract

Convolution is one of the most essential components of architectures used in computer vision. As machine learning moves towards reducing the expert bias and learning it from data, a natural next step seems to be learning convolution-like structures from scratch. This, however, has proven elusive. For example, current state-of-the-art architecture search algorithms use convolution as one of the existing modules rather than learning it from data. In an attempt to understand the inductive bias that gives rise to convolutions, we investigate minimum description length as a guiding principle and show that in some settings, it can indeed be indicative of the performance of architectures. To find architectures with small description length, we propose $\beta$-LASSO, a simple variant of LASSO algorithm that, when applied on fully-connected networks for image classification tasks, learns architectures with local connections and achieves state-of-the-art accuracies for training fully-connected nets on CIFAR-10 (85.19%), CIFAR-100 (59.56%) and SVHN (94.07%) bridging the gap between fully-connected and convolutional nets.

## 1   Introduction

Since its inception, machine learning has moved from inserting expert knowledge as explicit inductive bias toward general-purpose methods that learn these biases from data, a trend significantly accelerated by the advent of deep learning [16]. This trend is pronounced in areas such as computer vision [14], speech recognition [1], natural language processing [31] and computational biology [31]. In computer vision, for example, tools such as deformable part models [9], SIFT features [21] and Gabor filters [22] have all been replaced with deep convolutional architectures. Crucially, in certain cases, these general-purpose models learn similar biases to the ones present in the traditional, expert-designed tools.

Gabor filters in convolutional nets are a prime example of this phenomenon: convolutional networks learn Gabor-like filters in their first layer [38]. However, simply replacing the first convolutional layer by a Gabor filter worsens performance, showing that the learned parameters differ from the Gabor filter in a helpful way. A natural analogy can then be drawn between Gabor filters and the use of convolution itself: convolution is a 'hand-designed' bias which expresses local connectivity and weight sharing. Is it possible to *learn* these convolutional biases from scratch, the same way convolutional networks learn to express Gabor filters?

Reducing inductive bias requires more data, more computation, and larger models—consider replacing a convolutional network with a fully-connected one with the same expressive capacity. Therefore it is important to reduce the bias in a way that does not damage the efficiency significantly, keeping only the core bias which enables high performance. Is there a core inductive bias that gives rise to local connectivity and weight sharing when applied to images, enabling the success of convolutions? The answer to this question would enable the design of algorithms which can learn, e.g., to apply convolutions to images, and apply more appropriate biases for other types of data.

Current research in architecture search is an example of efforts to reduce inductive bias but often without explicitly substituting the inductive bias with a simpler one [41, 29, 34]. However, searching without a guidance makes the search computationally expensive. Consequently, the current techniques in architecture search are not able to find convolutional networks from scratch and only take convolution layer as a building block and focus on learning the interaction of the blocks.

**Related Work**   Previous work attempted to understand or improve the gap between convolutional and fully-connected networks. Perhaps the most related work is Urban et al. [36], titled *"Do deep convolutional nets really need to be deep and convolutional?"* (the abstract begins "Yes, they do."). Urban et al. [36] demonstrate empirically that even when trained with distillation techniques, fully-connected networks achieve subpar performance on CIFAR10, with a best-reported accuracy of 74.3%. Our work, however, suggests a different view, with results that significantly bridge this gap between fully-connected and convolutional networks. In another attempt, Mocanu et al. [23] proposed sparse evolutionary training, achieving 74.84% accuracy on CIFAR-10. Fernando et al. [10] also proposes an evolution-based approach but they only evaluate their approach on MNIST dataset. To the best of our knowledge, the highest accuracy achieved by fully-connected networks on CIFAR-10 is 78.62% [20], achieved through heavy data augmentation and pre-training with a zero-bias auto-encoder.

In order to understand the inductive bias of convolutional networks, d'Ascoli et al. [5] embedded convolutional networks in fully-connected architectures, finding that combinations of parameters expressing convolutions comprise regions which are difficult to arrive at using SGD. Other works have studied simplified versions of convolutions from a theoretical prospective [12, 3, 28]. Relatedly, motivated by compression, many recent works [11, 18, 8, 6] study sparse neural networks; studying their effectiveness on learning architectural bias from data would be an interesting direction.

Other recent interesting related work show variants of transformers are capable of succeeding in vision tasks and learning locality connected patterns [4, 2]. In order to do so, one needs to provide the pixel location as input which enables the attention mechanism to learn locality. Furthermore, it is not clear that in order to learn convolutions such complex architecture is required. Finally, in a parallel work Zhou et al. [40] proposed a method for meta-learning symmetries from data and showed that it possible to use such method to meta-learn convolutional architectures from synthetic data.

**Contributions:**   Our contributions in this paper are as follows:

- We introduce shallow (S-CONV) and deep (D-CONV) all-convolutional networks [33] with desirable properties for studying convolutions. Through systematic experiments on S-CONV and D-CONV and their locally connected and fully-connected counterparts, we make several observations about the role of depth, local connectivity and weight sharing (Section 2):
  - Local connectivity appears to have the greatest influence on performance.
  - The main benefit of depth appears to be efficiency in terms of memory and computation. Consequently, training shallow architectures with many more parameters for a long time would compensate most of the lost performance due to lack of depth.
  - The benefit of depth diminishes even further without weight-sharing.
- We look at Minimum Description Length (MDL) as a guiding principle to what architectures generalize better (Section 3):
  - Showing that MDL can be bounded by number of parameters, we argue and demonstrate empirically that architecture families that need fewer parameters to fit a training set a certain degree tend to generalize better in the over-parameterized regime.
  - We prove an MDL-based generalization bound for architectures search which suggests that the sparsity of the found architecture has great effect on generalization. However, weight sharing is only effective if it has a simple structure.
- Inspired by MDL, we propose a training algorithm $\beta$-LASSO , a variant of LASSO with a more aggressive soft-thresholding to find architectures with few parameters and hence, a small description length. We present the following empirical findings for $\beta$-LASSO (Section 4):
  - $\beta$-LASSO achieves state-of-the-art results on training fully connected networks on CIFAR10, CIFAR-100 and SVHN tasks. The results are on par with the reported performance of multi-layer convolutional networks around year 2013 [15, 37]. Moreover, unlike convolutional networks, these results are *invariant to permuting pixels*.

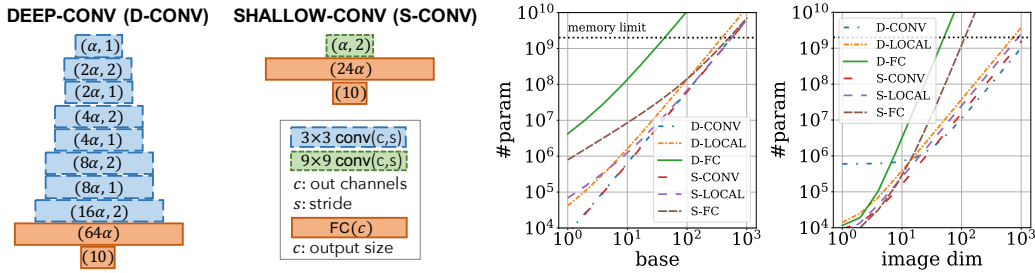

Figure 1: **D-CONV and S-CONV architectures and their scaling**: Left panel shows the architectures. Each convolution or fully-connected layer (except the last one) has batch-normalization followed by ReLU activation. The right panel indicates how the number of parameters in each architecture and their corresponding locally and fully-connected scales with respect to the number of base channels (shown by $\alpha$ in the left panel) and image dimension. The dotted black line is the maximum model size for training using 16-bits on a V100 GPU.

– We show that the learned networks have fewer parameters than their locally connected counterparts. By visualizing the filters, we observe that $\beta$-LASSO has indeed learned local connectivity but it has also learned to sample more sparsely in a local neighborhood to increase the receptive field while keeping the number of parameters low.

– Somewhat related to the main goal of the paper, we trained ResNet18 with different kernel sizes using $\beta$-LASSO and we observed that for all kernel sizes, $\beta$-LASSO improves over SGD on CIFAR10, CIFAR-100 and SVHN datasets.

## 2 Disentangling Depth, Weight sharing and Local connectivity

To study the inductive bias of convolutions, we take a similar approach to d'Ascoli et al. [5], examining two large hypothesis classes that encompass convolutional networks: locally-connected and fully-connected networks. Given a convolutional network, its corresponding locally-connected version features the same connectivity, but eliminates weight sharing. The corresponding fully-connected network then adds connections between all nodes in adjacent layers. Functions expressible by the fully-connected networks encompass those of locally-connected networks, which encompass convolutional networks.

One challenge in studying the inductive bias of convolutions is that the existence of other components such as pooling and residual connections makes it difficult to isolate the effect of convolutions in modern architectures. One solution is to simply remove those from the current architectures. However, that would result is a considerable performance loss since the other design choices of the architecture family were optimized with those components. Alternatively, one can construct an all-convolutional network with a desirable performance. Springenberg et al. [33] have proposed a few all-convolutional architectures with favorable performance. Unfortunately, these models cannot be used for studying the convolution since fully-connected networks that can represent such architectures are too large. Another way to resolve this is by scaling down the number of parameters in the conventional architecture which unfortunately degrades the performance significantly. To this end, below we propose two all-convolutional networks to overcome the discussed issues.

### 2.1 Introducing D-CONV and S-CONV for Studying Convolutions

In this work, we propose D-CONV and S-CONV, two all-convolutional networks that perform relatively well on image classification tasks while also enjoying desirable scaling with respect to the number of channels in the corresponding convolutional network and input image size. As shown in the left panel of Figure 1, D-CONV has 8 convolutional layers followed by two fully-connected layers. However, S-CONV has only one convolutional layer which is followed by two fully-connected layers. The right panel in Figure 1 shows how the number of parameters of these models and their corresponding locally-connected (D-LOCAL, S-LOCAL) and fully-connected (D-FC, S-FC) networks scale with respect to the number of base channels (channels in the first convolutional layer) and the input image size. D-FC has the highest number of parameters given the same number of base channels, which

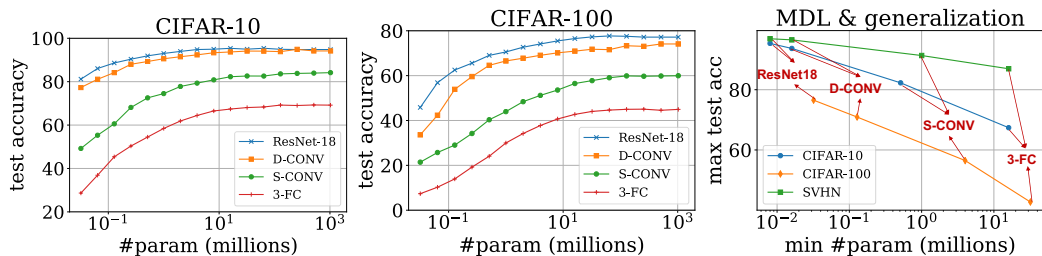

Figure 2: **Performance scaling of different architectures.** The left and middle panels show the test accuracy of four architectures trained on CIFAR-10 and CIFAR-100 datasets. See the Appendix for SVHN plot and experiment details. The right panel shows the maximum test accuracy of architectures in the over-parameterized regime against the minimum number of parameters they need in order to achieve a certain training accuracy that is fixed for each dataset.

means the largest D-CONV that can be studied will not have many based channels. On the other hand S-FC has a better scaling which allows us to have more base channels in the corresponding S-CONV. The other interesting observation is that the number of parameters in fully-connected networks depends on the fourth power of the input image dimension (e.g. 32 for CIFAR-10). However, the local and convolutional networks have a quadratic dependency on the image dimension. Note that the quadratic dependency of D-CONV and S-CONV to the image size is due to of lack of global pooling.

Figure 2 compares the performance of D-CONV and S-CONV against ResNet18 [13], and a 3-layer (2 hidden layers) fully-connected network with equal number of hidden units in each hidden layers denoted by 3-FC. It is clear that on all tasks, the performance of D-CONV is comparable but slightly worse than ResNet18 which is expected. Moreover, S-CONV has a performance which is significantly better than fully-connected networks but worse than D-CONV .

| Models | #param (M) | | CIFAR10 | | CIFAR100 | | SVHN | |
| | orig. | FC-emb. | 400 ep. | 4000 ep. | 400 ep. | 4000 ep. | 400 ep. | 4000 ep. |
|---|---|---|---|---|---|---|---|---|
| D-CONV | 1.45 | 256 | 88.84 | 89.79 | 63.73 | 62.26 | 95.65 | 95.86 |
| D-LOCAL | 3.42 | 256 | 86.13 | 86.07 | 58.58 | 55.71 | 95.71 | 95.85 |
| D-FC | 256 | 256 | 64.78 | 63.62 | 36.61 | 35.51 | 92.52 | 90.97 |
| S-CONV | 138 | 256 | 84.14 | 87.05 | 59.48 | 62.51 | 92.34 | 93.38 |
| S-LOCAL | 147 | 256 | 81.52 | 85.86 | 56.64 | 62.03 | 92.51 | 93.98 |
| S-FC | 256 | 256 | 72.77 | 78.63 | 47.72 | 51.43 | 88.64 | 91.80 |
| 3-FC | 256 | 256 | 69.19 | 75.12 | 44.95 | 50.75 | 85.98 | 86.02 |

Table 1: Performance of D-CONV, S-CONV and their locally and fully-connected counterparts. The results indicate that in this regime, the performance of S-LOCAL is comparable to that of D-LOCAL and S-CONV . Please refer to Appendix for details of the training procedure.

## 2.2 Empirical Investigation

In this section, we disentangle the effect of depth, weight sharing and local connectivity. Table 1 reports the test accuracy of D-CONV, S-CONV, their counterparts and 3-FC on three datasets. For each architecture, the base channels is chosen such that the corresponding fully-connected network has roughly 256 million parameters. First note that with that constraint, deep convolutional and locally connected architectures have much smaller number of parameters compare to others. Moreover, for both deep and shallow architectures, there is not a considerable difference in the number of parameters of the convolutional and locally connected networks. Please refer to Figure 1 for the scaling of each architecture. For each experiment, the results for training 400 and 4000 epochs are reported. Several observations can be made from Table 1:

1. **Importance of locality**: For both deep and shallow architectures and across all three datasets, the gap between a locally connected network and its corresponding fully-connected version is much higher than that between convolutional and locally-connected networks. That suggests that the main benefit of convolutions comes from local connectivity.

2. **Shallow architectures eventually catch up to deep ones (mostly)**: While training longer for deep architectures does not seem to improve the performance, it significantly does so in the case of shallow architectures across all datasets. As a result, the gap between deep and shallow architectures shrinks significantly after training for 4000 epochs.

3. **Without weight sharing, the benefit of depth disappears**: S-FC outperforms D-FC in all experiments. Moreover, when training for 4000 epochs, none of D-LOCAL and S-LOCAL have clear advantage over each other.

4. **Structure of fully-connected network matters**: S-FC outperforms 3-FC and D-FC in all experiments by a considerable margin. Even more interesting is that S-FC and 3-FC have the same number of parameters and depth but S-FC has much more hidden units in the first layer compare to 3-FC. Please see the appendix for the exact details.

The results in Table 1 suggests that S-CONV performs comparably to D-CONV and the gap between S-CONV and S-LOCAL is negligible compare to the gap between S-LOCAL and S-FC . Therefore, in the rest of the paper, we try to bridge the gap between the performance of S-FC and S-LOCAL .

## 3 Minimum Description Length as a Guiding Principle

In this section, we take a short break from experiments and look at the minimum description length as a way to explain the differences in the performance of architectures and a guiding principle for finding models that generalize well. Consider the supervised learning setup where given an input data $\mathbf{x} \in \mathcal{X}$, we want to predict a label $\mathbf{y} \in \mathcal{Y}$ under the common assumption that the pair $(\mathbf{x}, \mathbf{y})$ are generated from a distribution $\mathcal{D}$. Let $\mathcal{S} = \{(\mathbf{x}_1, \mathbf{y}_n) \ldots, (\mathbf{x}_m, \mathbf{y}_m)\}$ be a training set sampled i.i.d from $\mathcal{D}$ and $\ell : \mathcal{Y} \times \mathcal{Y} \to R$ be a loss function. The task is then to learn a function/hypothesis $h : \mathcal{X} \to \mathcal{Y}$ with low population loss $L_{\mathcal{D}}(h) = \mathbb{E}_{(\mathbf{x}, \mathbf{y}) \sim \mathcal{D}} [\ell(h(\mathbf{x}), \mathbf{y})]$ also known as generalization error. In order to find the hypothesis $h$, we start by picking a hypothesis class $\mathcal{H}$ (eg. linear classifiers, neural networks of a certain family, etc.) and a learning algorithm $\mathcal{A}$. Finally, $\mathcal{A}(\mathcal{S}, \mathcal{H})$ returns a hypothesis $h$ which will be used for prediction. The learning algorithm usually finds a hypothesis among the ones that have low sample loss $L_{\mathcal{S}} = \frac{1}{m} \sum_{(\mathbf{x}, \mathbf{y}) \in \mathcal{S}} \ell(h(\mathbf{x}), \mathbf{y})$.

A fundamental question in learning concerns the *generalization gap* between the training error and the generalization error. One way to think about generalization is to think of a hypothesis as an explanation for association of label $\mathbf{y}$ to input data $\mathbf{x}$ (eg. given a picture and the label "dog", the hypothesis is an explanation of what makes this picture, a picture of a dog). The Occam's razor principle provides an intuitive way of thinking about generalization of a hypothesis [32]:

> *A short explanation tends to be more valid than a long explanation.*

The above philosophical message can indeed be formalized as follows:

**Theorem 1.** *(32, Theorem 7.7) Let $\mathcal{H}$ be a hypothesis class and $d : \mathcal{H} \to \{0, 1\}^*$ be a prefix-free description language for $\mathcal{H}$. Then for any distribution $\mathcal{D}$, sample size $m$, and probability $\delta > 0$, with probability $1 - \delta$ over the choice of $\mathcal{S} \sim \mathcal{D}^m$ we have that for any $h \in \mathcal{H}$*

$$L_{\mathcal{D}}(h) \leq L_{\mathcal{S}}(h) + \sqrt{\frac{|d(h)| + \log(2/\delta)}{2m}} \qquad (1)$$

*where $|d(h)|$ is the length of $d(h)$.*

The above theorem connects the generalization gap of a hypothesis to its description length using a prefix-free description language, i.e. for any two distinct $h$ and $h'$, $d(h)$ is not a prefix of $d(h')$. Importantly, the description language should be chosen before observing the training set. The simplest form of a prefix free description language is the bit representation of the parameters. If a model has $n$ parameters each of which stored in $b$ bits, then the total number of bits to describe a model is $nb$. Furthermore, note that this is prefix-free because all hypotheses have the same description length. According to this language, the generalization gap is simply controlled by the number of parameters.

Empirical investigations on generalization of over-parameterized models suggest that the number of parameters is a very loose upper bound on the capacity [17, 27, 39, 7, 25] (or in MDL language, it is not a compact encoding). However, it is possible that the number of parameters is a compact encoding in the under-parameterized regime but some other encoding becomes optimal in the over-parameterized regime. One empirical observation about many neural network architectures is that with a well-chosen approach to scale the number of parameters, one can expect that over-parametrization does not make a superior architecture inferior. While this is not always the case, it might be enough to distinguish architectures that have very different inductive biases. For example, the left two panels of Figure 2 show the performance plots of different architectures do not cross each other across the scale. This was also the central observation used in architecture search by Tan and Le [34].

When the ordering of architectures with the same number of parameters based on their test performance is the same for any number of parameters (similar to the left two panels of Figure 2), the performance in under-parameterized regime can be indicative of the performance in over-parameterized regime and hence if an architecture can fit the training data with fewer number of parameters, that most likely translate to the superiority of the architecture in the over-parameterized regime. In the right panel of Figure 2, we can see that it is indeed the case for all 4 architectures and 3 datasets that we study in this work. ResNet18 requires least number of parameters to fit any dataset while the next ones are D-CONV, S-CONV and 3-FC respectively and this is the exact order in terms of generalization performance in over-parameterized regime where models have around 1 billion parameters.

**MDL-based generalization bound for architecture search**  Theorem 1 gives a bound on the description length of a hypothesis and we discussed how it can be bounded by number of parameters. What if the learning algorithm searches over many architectures and finds one with small number of parameters? In this case, the active parameters can be denoted as the parameters with non-zero values and the parameter sharing can be modeled as parameters with the same value. Does the performance then only depend on the number of parameters of the final architecture? How does the search space come to the picture and how does weight sharing affect the performance? Let $b$ be the number of bits used to present each parameter (usually 16 or 32) and $n$ be the maximum number of allowed parameters in the architectures found by the architecture algorithm. Also, let $k$ be the number of parameters in the architecture found by the architecture search algorithm. The next theorem shows how the generalization gap can be bounded in this case.

**Theorem 2.** *Let $\mathbb{R}_b \subset \mathbb{R}$ be a $b$-bit presentation of real numbers ($|\mathbb{R}_b| = 2^b$), $\mathcal{F}$ be a class of parameterized functions $f_\mathbf{w}$ where $\mathbf{w} \in \mathbb{R}_b^n$, $\mathcal{G} = \{g : [n] \to [k] | k \leq n\}$ be all possible mappings from numbers $1 \ldots n$ to $1 \ldots k$ for any $k \leq n$ and $d : \mathcal{G} \to \{0,1\}^*$ a be prefix-free description language for $\mathcal{G}$. For any distribution $\mathcal{D}$, sample size $m$, and $\delta > 0$, with probability $1 - \delta$ over the choice of $\mathcal{S} \sim \mathcal{D}^m$, for any $f_\mathbf{w} \in \mathcal{F}$:*

$$L_\mathcal{D}(f_\mathbf{w}) \leq L_\mathcal{S}(f_\mathbf{w}) + \sqrt{\frac{kb + |d(g)| + \log(2n/\delta)}{2m}} \tag{2}$$

*where $|d(g)|$ is the length of $d(g)$ and $w_i = p_{g(i)}$ for some $p \in \mathbb{R}_b^k$. Moreover, there exists a prefix free language $d$ such that for any $g \in \mathcal{G}$, $|d(g)| \leq \|\mathbf{w}\|_0 \log(kn) + 2\log(n)$.*

The proof is given in the appendix. The above theorem shows that the capacity of the learned model is bounded by two terms: number bits to present the parameters of the learned architecture ($kb$) and the description length of the weight sharing ($d(g)$). This is rewarding for finding architectures with small number of non-zero weights because in that case the generalization gap would mostly depend on the number of non-zero parameters (using $d(g) \leq \|\mathbf{w}\|_0 \log(kn) + 2\log(n)$). However, in the case of weight sharing, the picture is very different. The bound suggests that the generalization gap depends on the description length of the weight sharing and that could in worst case depend on the number of non-zero parameters if there is no structure. Otherwise, when weight-sharing is structured, it can be encoded more efficiently and the generalization could potentially only depend on the number of parameters of the final model found by the architecture search. This suggests that simply encouraging parameter values to be close to each other does not improve generalization unless there is a structure in place. Therefore, we leave this for future work and focus on learning networks with small number of non-zero weights in the rest of the paper which seemed to be the most important element based on our empirical investigation in Section 2.2.

---
**Algorithm 1** $\beta$-LASSO
---
**Parameters:** $f(\theta)$: stochastic objective function with parameters $\theta$, $\theta_0$: the initial parameter vector, $\lambda$: coefficient of $\ell_1$ regularizer, $\beta$: threshold coefficient, $\eta$: learning rate, $\tau$: number of updates

  1: **for** $t = 1$ **to** $\tau$ **do**
  2:      $\theta_t \leftarrow \theta_{t-1} - \eta(\nabla_\theta f(\theta_{t-1}) + \lambda \operatorname{sign}(\theta_{t-1}))$
  3:      $\theta_t \leftarrow \theta_t \left(|\theta_t| \geq \beta\lambda\right)_+$
  4: **Return** $\theta_t$

---

## 4 $\beta$-LASSO : Learning Local Connectivity from Scratch

In the previous section, we discussed that if a learning algorithm can find an architecture with a small number of non-zero weights, then the generalization gap of that architecture would mostly depend on the number of non-zero weights and the dependence on the number of original parameters would be only logarithmic. This is very encouraging. On the other hand, we know from our empirical investigation in Section 2.2 that locally connected networks perform considerably better than fully-connected ones. Is it possible to find locally connected networks by simply encouraging sparse connections when training on images? Could such networks bridge the gap between performance of fully-connected and convolutional networks?

We are interested in architectures with small number of parameters. In order to achieve this, we try the simplest form of encouraging the sparsity by adding an $\ell_1$ regularizer. In particular, we propose $\beta$-LASSO , a simple algorithm that is very similar to LASSO [35] except it has an extra parameter that allows for more aggressive soft thresholding. The algorithm is shown in Algorithm 1.

### 4.1 Training fully-connected Networks

Table 2 compares the performance of S-FC trained with $\beta$-LASSO to state-of-the-art methods in training fully-connected networks. The results show a significant improvement over previous work even considering complex methods such distillation or pretraining. Moreover, there is only very small gap between performance of S-FC trained with $\beta$-LASSO and its locally connected and convolutional counterparts. However, note that unlike S-CONV and S-LOCAL, the performance of S-FC is invariant to permuting the pixels which is a considerable advantage when little is known about the structure of data. To put these results to perspective, these accuracies are on par with best results for convolutional networks in 2013 [15, 37]. We fix $\beta$ in the experiments but tune the regularization parameter $\lambda$ for each datasets. We also observed that having higher $\lambda$ for the layer that corresponds to the convolution, improves the performance which is aligned with our understanding that such a layer could benefit the most from sparsity.

| Model | Training Method | CIFAR-10 | CIFAR-100 | SVHN |
|---|---|---|---|---|
| S-CONV | SGD | 87.05 | 62.51 | 93.38 |
| S-LOCAL | SGD | 85.86 | 62.03 | 93.98 |
| MLP [26] | SGD (no Augmentation) | 58.1 | - | 84.3 |
| MLP [24] | Adam/RMSProp | 72.2 | 39.3 | - |
| MLP [23] | SET(Sparse Evolutionary Training) | 74.84 | - | - |
| MLP [36] | deep convolutional teacher | 74.3 | - | - |
| MLP [20] | unsupervised pretraining with ZAE | 78.62 | - | - |
| MLP (3-FC) | SGD | 75.12 | 50.75 | 86.02 |
| MLP (S-FC) | SGD | 78.63 | 51.43 | 91.80 |
| MLP (S-FC) | $\beta$-LASSO ($\beta = 0$) | 82.45 | 55.58 | 93.80 |
| MLP (S-FC) | $\beta$-LASSO ($\beta = 1$) | 82.52 | 55.96 | 93.66 |
| MLP (S-FC) | $\beta$-LASSO ($\beta = 50$) | **85.19** | **59.56** | **94.07** |

Table 2: Comparing the performance S-FC trained with $\beta$-LASSO to other methods for training fully-connected networks. Please see the appendix for details of the training procedure.

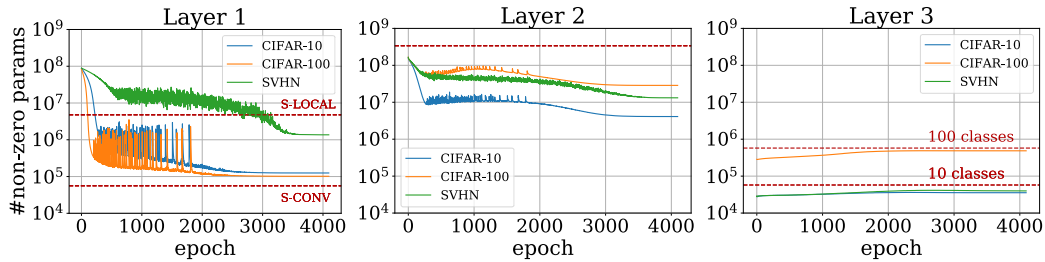

Figure 3: **Number of non-zero parameters in different layers of S-FC trained with $\beta$-LASSO** . The left panel indicates that even though the first layers of S-CONV and S-LOCAL have much fewer number of parameters (red dashed lines) compare to S-FC , the number of non-zero parameters in trained S-FC is between that of S-LOCAL and S-CONV . The middle and right panels show the same quantity for the second and third layers of S-FC which have the same parametrization as S-CONV and S-LOCAL . The plots suggest that $\beta$-LASSO encourages the weights of the second layer to be much sparser but the last layer remains dense.

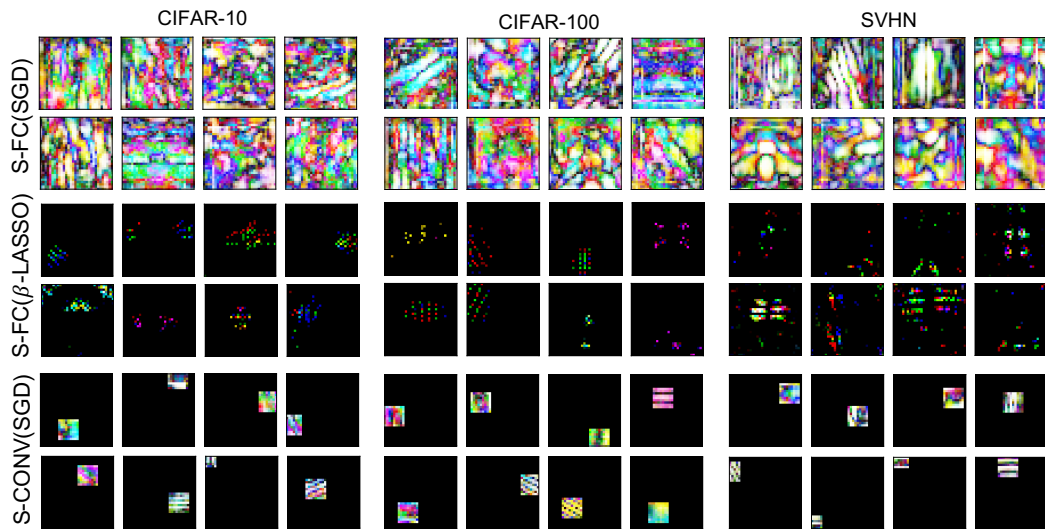

Figure 4: Comparing the layers of the architectures when trained with $\beta$-LASSO and when trained with SGD. The filters learned by SGD training of S-FC are dense but locally correlated. However, the filters learned when trained with $\beta$-LASSO are locally connected in a similar way to S-LOCAL . Furthermore, it seems that the network has learned that nearby pixels have similar information and therefore to get more information while remaining local, it learns to look at a sparse sampling of a local neighborhood.

Furthermore, to see if $\beta$-LASSO succeeds in learning architectures that are as sparse as S-LOCAL, we measure the number of non-zero weights in each layer separately. The results are shown in Figure 3. The left panel corresponds to the first layer which is convolutional in S-CONV and locally connected in S-LOCAL but fully-connected in S-FC . As you can see, the solution found by $\beta$-LASSO has less nonzero parameters than the number of parameters in S-LOCAL and has only slightly more parameters than S-CONV even though S-CONV is benefiting from weight-sharing. The middle and left plots show that in other layers, the solution found by $\beta$-LASSO is still sparse but less so in the final layer.

Our further investigation into the learned filters resulted in surprising results that is shown in Figure 4. S-FC trained with SGD learned filters that are very dense but with locally correlated values. However, the same network trained with $\beta$-LASSO learns a completely different set of filters. The filters learned by $\beta$-LASSO are sparse and locally connected in a similar fashion to S-LOCAL filters. Moreover, it appears that the networks trained with $\beta$-LASSO have learned that immediate pixels have little new information. In order to benefit from larger receptive fields while remaining local and sparse, these networks have learned filters that seem like a sparse sampling of a local neighborhood in the image. This encouraging finding validates that by using a learning algorithm with better inductive bias, one can start from an architecture without much bias and learn it from data.

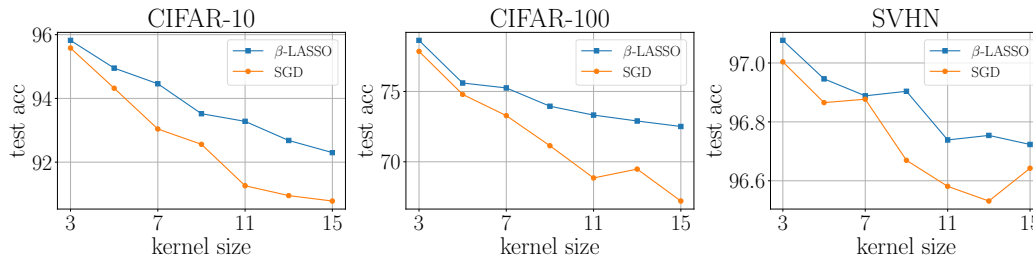

Figure 5: Performance of ResNet18 trained with different kernel sizes.

## 4.2 Training Convolutional Networks with Larger Kernel Size

Slightly deviating from our main goal, we tried training ResNet18 with different kernel sizes using $\beta$-LASSO and compare it with SGD. Figure 5 shows that $\beta$-LASSO improves over SGD for all kernel sizes across all datasets. The improvement is predictably more so when kernel size is large since $\beta$-LASSO would learn to adjust it automatically. These results again confirm that $\beta$-LASSO can be used in many different settings to adoptively learn the structure.

## 5 Discussion and Future Work

In this work, we studied the inductive bias of convolutional networks through empirical investigations and MDL theory. We proposed a simple algorithm called $\beta$-LASSO that significantly improves training of fully-connected networks. $\beta$-LASSO does not have any specific inductive bias related to images. For example, permuting all pixels in our experiments does not change the performance of $\beta$-LASSO. It is therefore interesting to see if $\beta$-LASSO can be used in other contexts such as natural language processing or in domains such as computational biology where the data is structured but our knowledge of the structure of the data is much more limited compare to computer vision. Another promising direction is to improve efficiency of $\beta$-LASSO to benefit from sparsity for faster computation and less memory usage. The current implementation does not benefit from sparsity in terms of memory or computation. In order to scale up $\beta$-LASSO to be able to handle larger networks and input data dimensions, these barriers should be removed. Finally, we want to emphasize general purpose algorithms such as $\beta$-LASSO that are able to learn the structure become even more promising as training larger models become more accessible.

## Broader Impact

This work studies inductive bias of convolutions and proposes an algorithm to learn local connectivity from data. Since this work aims to improve a fundamental aspect of deep learning, we believe it does not have immediate negative societal consequences. In the long term, making progress on general purpose algorithms allows machine learning engineers to spend less time on architecture design and more on other aspects of learning algorithms.

## Acknowledgments and Disclosure of Funding

We thank Sanjeev Arora, Ethan Dyer, Guy Gur-Ari, Aitor Lewkowycz and Yuhuai Wu for many fruitful discussions and Vaishnavh Nagarajan and Vinay Ramasesh for their useful comments. This work was funded by Google.

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
