[Supplementary Material]

# A  Experimental Setup

We used Caliban [30] to manage all experiments in a reproducible environment in Google Cloud's AI Platform. In all experiments, we used 16-bit operations on V100 GPUs except Batch Normalization which was kept at 32-bit.

## A.1  Architectures

The ResNet18 architecture used in our experiments is the same its original version introduced in He et al. [13] for training ImageNet dataset except that the first convolution layer has kernel size 3 by 3 and it is not followed by a max-pooling layer. This changes are made to adjust the architecture for smaller image sizes used in our experiments. 3-FC architecture is a fully connected network with two hidden layers having the same number of hidden units with ReLU activation and Batch Normalization. Finally, please see Tables 3 and 4 for details of D-CONV, S-CONV and their locally and fully connected counterparts.

| Layer | Description | #param | | |
|-------|-------------|--------|--------|--------|
| | | D-CONV | D-LOCAL | D-FC |
| conv1 | CONV3x3$(\alpha, 1)$ | $9 \times 3\alpha$ | $9 \times 3s^2\alpha$ | $3s^4\alpha$ |
| conv2 | CONV3x3$(2\alpha, 2)$ | $9 \times 2^1\alpha^2$ | $9 \times s^2\alpha^2/2$ | $s^4\alpha^2/2$ |
| conv3 | CONV3x3$(2\alpha, 1)$ | $9 \times 2^2\alpha^2$ | $9 \times s^2\alpha^2/4$ | $s^4\alpha^2/4$ |
| conv4 | CONV3x3$(4\alpha, 2)$ | $9 \times 2^3\alpha^2$ | $9 \times s^2\alpha^2/8$ | $s^4\alpha^2/8$ |
| conv5 | CONV3x3$(4\alpha, 1)$ | $9 \times 2^4\alpha^2$ | $9 \times s^2\alpha^2/16$ | $s^4\alpha^2/16$ |
| conv6 | CONV3x3$(8\alpha, 2)$ | $9 \times 2^5\alpha^2$ | $9 \times s^2\alpha^2/32$ | $s^4\alpha^2/32$ |
| conv7 | CONV3x3$(8\alpha, 1)$ | $9 \times 2^6\alpha^2$ | $9 \times s^2\alpha^2/64$ | $s^4\alpha^2/64$ |
| conv8 | CONV3x3$(16\alpha, 2)$ | $9 \times 2^7\alpha^2$ | $9 \times s^2\alpha^2/128$ | $s^4\alpha^2/128$ |
| fc1 | FC$(64\alpha)$ | $4s^2\alpha^2$ | $4s^2\alpha^2$ | $4s^2\alpha^2$ |
| fc2 | FC(c) | $64c\alpha$ | $64c\alpha$ | $64c\alpha$ |
| Total | D-CONV$(\alpha)$ | $\approx (9 \times 2^8 + 4s^2)\alpha^2$ $+(27 + 64c)\alpha$ | $\approx 13s^2\alpha^2$ $+(27s^2 + 64c)\alpha$ | $\approx (s^4 + 4s^2)\alpha^2$ $+(3s^4 + 64c)\alpha$ |

Table 3: D-CONV and its counterparts. $\alpha$ denotes the number of base channels which determines the total number of parameters of the architecture. The convolutional modules in D-CONV turn into locally connected and fully connected modules for D-LOCAL and D-FC respectively.

## A.2  Hyperparameters

All architectures in this paper are trained with Cosine Annealing learning rate schedule with initial learning rate $0.1$, batch size $512$ and data augmentation proposed in Lim et al. [19]. We looked at learning rates $0.01$ and $1$ as well but observed that $0.1$ works well across experiments. For models trained with SGD, we tried values $0$ and $0.9$ for the momentum, and $0$, $10^{-4}$, $2 \times 10^{-4}$, $5 \times 10^{-4}$, $10^{-3}$ for the weight decay. As for $\beta$-LASSO, we did not use momentum, set $\beta = 50$ and tried $10^{-6}$, $2 \times 10^{-6}$, $5 \times 10^{-6}$, $10^{-5}$ and $2 \times 10^{-5}$ for the choice of $\ell_1$ regularization for layers that correspond to convolutional and fully connected layers separately. Moreover, for all models, we tried both adding dropout to the last two fully connected layers and not adding dropout to any layers. Models in Table 2 are trained for 4000 epochs. All other models in this paper are trained for 400 epochs unless mentioned otherwise. For each experiment, picked models with highest validation accuracy among these choices of hyper-parameters.

# B  Proof of Theorem 2

In this section, we restate and prove Theorem 2.

**Theorem 2.** *Let $\mathbb{R}_b \subset \mathbb{R}$ be a b-bit presentation of real numbers ($|\mathbb{R}_b| = 2^b$), $\mathcal{F}$ be a class of parameterized functions $f_{\mathbf{w}}$ where $\mathbf{w} \in \mathbb{R}_b^n$, $\mathcal{G} = \{g : [n] \to [k] | k \leq n\}$ be all possible mappings from numbers $1 \ldots n$ to $1 \ldots k$ for any $k \leq n$ and $d : \mathcal{G} \to \{0, 1\}^*$ a be prefix-free description*

| Layer | Description | #param | | |
|-------|-------------|--------|--------|------|
| | | S-CONV | S-LOCAL | S-FC |
| conv1 | CONV9x9$(\alpha, 2)$ | $81 \times 3\alpha$ | $81 \times 3s^2\alpha/4$ | $3s^4\alpha/4$ |
| fc1 | FC$(24\alpha)$ | $6s^2\alpha^2$ | $6s^2\alpha^2$ | $6s^2\alpha^2$ |
| fc2 | FC(c) | $24c\alpha$ | $24c\alpha$ | $24c\alpha$ |
| Total | S-CONV$(\alpha)$ | $6s^2\alpha^2$ $+(243 + 24c)\alpha$ | $6s^2\alpha^2$ $+(243s^2/4 + 24c)\alpha$ | $6s^2\alpha^2$ $+(3s^4/4 + 24c)\alpha$ |

Table 4: S-CONV and its counterparts. $\alpha$ denotes the number of base channels which determines the total number of parameters of the architecture. The convolutional modules in S-CONV turn into locally connected and fully connected modules for S-LOCAL and S-FC respectively.

*language for $\mathcal{G}$. For any distribution $\mathcal{D}$, sample size $m$, and $\delta > 0$, with probability $1 - \delta$ over the choice of $\mathcal{S} \sim \mathcal{D}^m$, for any $f_{\mathbf{w}} \in \mathcal{F}$:*

$$L_{\mathcal{D}}(f_{\mathbf{w}}) \leq L_{\mathcal{S}}(f_{\mathbf{w}}) + \sqrt{\frac{kb + |d(g)| + \log(2n/\delta)}{2m}} \tag{2}$$

*where $|d(g)|$ is the length of $d(g)$ and $w_i = p_{g(i)}$ for some $p \in \mathbb{R}_b^k$. Moreover, there exists a prefix free language $d$ such that for any $g \in \mathcal{G}$, $|d(g)| \leq \|\mathbf{w}\|_0 \log(kn) + 2\log(n)$.*

*Proof.* We build our proof based on theorem 1 and construct a prefix-free description language for $f_{\mathbf{w}}$. We use the first $\log(n)$ bits to encode $k \leq n$ which is the dimension of parameters. The next $kb$ bits then would be used to store the values of parameters. Note that so far, the encoding has varied length based on $k$ but it is prefix free because for any two dimensions $k_1 \neq k_2$, the first $\log(n)$ bits are different. Finally, we add the encoding for $g$ which takes $d(g)$ bit based on theorem statement. After adding $d$ the encoding remains prefix-free. The reason is that for any two $f_{\mathbf{w}}$, if they have different number of parameters, the first $\log(n)$ bits will be different. Otherwise, if they have the exact same number of parameters and the same parameter values, the length of the encoding before $d(g)$ will be exactly $\log(n) + kb$ and since $d$ is prefix free, the whole encoding remains prefix-free.

Next we construct a prefix-free description $d$ such that for any $g \in \mathcal{G}$, $d(g) \leq \|\mathbf{w}\|_0 \log(n)$. Instead of assigning all weights to parameters, it is enough to specify the weights that are assigned to non-zero parameters and the rest of the weights will then be assigned to the parameter with value zero. Similar to before, we use the first $\log(n)$ bits to specify the number of non-zero weights, i.e. $\|\mathbf{w}\|_0$ and the next $\log(n)$ bits to specify the number of parameters $k$. This again helps to keep the encoding prefix-free. Finally for each non-zero weight, we use $\log(n)$ bits to identify the index and $\log(k)$ to specify the index of its corresponding parameter. Therefore, the total length will be $\|\mathbf{w}\|_0 \log(kn) + 2\log(n)$ and this completes the proof. $\square$

## C   Supplementary Figures

Figure 6: **Performance scaling of different architectures.** The top row shows the training accuracy and the bottom row shows test accuracy

(a) CIFAR-10        (b) CIFAR-100        (c) SVHN

Figure 7: First layer filters of s-FC trained with SGD. This filters are chosen randomly among all filters with at least 20 non-zero values.

(a) CIFAR-10&emsp;&emsp;&emsp;&emsp;(b) CIFAR-100&emsp;&emsp;&emsp;&emsp;(c) SVHN

Figure 8: First layer filters of s-FC trained with $\beta$-LASSO. This filters are chosen randomly among all filters with at least 20 non-zero values.

|                |                 |            |
|:--------------:|:---------------:|:----------:|
| (a) CIFAR-10   | (b) CIFAR-100   | (c) SVHN   |

Figure 9: First layer filters of S-LOCAL trained with SGD. This filters are chosen randomly among all filters with at least 20 non-zero values.