[Reviews · NeurIPS 2020]

Review 1

Summary and Contributions: The paper offers an interesting investigation of the importance of inductive biases such as convolutions. The importance of local connections are explored, leading to the minimum description length as a measure. Further the theoretical backing though limited is interesting. Experimental results show that beta lasso is an effective technique in limiting the number of non zeros leading to improved performance.

Strengths: The paper offers a good walkthrough and development of its ideas. From the motivating experiments, to some theoretical backing, to developing and presenting the simple algorithms. Though the study overall is simple and reinforces known intuitions, it serves as a good reference for future work with concrete evidence. Experiments seems sufficient for this level of work, though investigations into skip connections through resents or densenets would have been interesting and necessary for better adoption.

Weaknesses: The major concern is that the paper only presents results on image or visual data. The motivation was to find inductive biases in other kinds of data since beta lasso would encourage the correct weights to be non-zero leading to found structure within the data. However this was not the case, no evidence is presented which is disappointing. Any NLP studies, audio studies etc would’ve helped the method to stand out in terms of the primary motivation of this study. This is the primary and major concern of this reviewer.

Correctness: Seems correct

Clarity: Yes

Relation to Prior Work: Yes

Reproducibility: Yes

Additional Feedback: I have read all reviewing material including the rebuttal and have decided to upgrade my score to a marginal accept.


Review 2

Summary and Contributions: This paper looks at the impact of sparsity in training neural networks. They begin by empirically analysing two neural network architectures (S-CONV and S-LOCAL) and study the impact of local vs global connectivity. Then they repurpose an MDL Occam bound to provide a generalization bound for neural networks, and inspired my MDL as a capacity control, introduce the \beta-LASSO algorithm to train neural networks. The algorithm is benchmarked on CIFAR-10, CIFAR-100 and SVHN image classification datasets.

Strengths: - The paper is easy to read. - The local connectivity experiments (Section 2) are a good approximation of disentangling the effect of various components within the convolutional architecture. - Experimental results are competitive.

Weaknesses: - I am not a fan in general of introducing new CNN architectures (S-FC, S-CONV etc.), and the reason being that they remain an isolated academic exercise, with little impact to research in applied domains (computer vision or natural language processing). The reason for this is that the base neural network architectures are mostly fixed after years of refinement in the specific area of architecture selection, therefore it is very unlikely that any of these target communities will readily adapt to these new architectures, especially considering that while they perform reasonably well, they do not provide state-of-the-art results, and in fact, underperform compared to (more on this later). - The above point being said, I do agree that there is a lot of value in learning sparser variants of these architectures, and providing rigorous regularizers for deep neural network models. This paper does demonstrate benefits of \beta-LASSO over vanilla SGD on ResNet-18 for various kernel sizes, but that's about the only conventional architecture it has been tested on. The remainder of comparisons are done on the architectures introduced by the authors, which, while interesting, has limited applicability: there is no evidence to suggest that S-FC and S-CONV generalize to tougher problems (ImageNet, CUB-2011, etc.). - \beta-LASSO and Table 2: The authors suggest that \beta-LASSO is a variant of LASSO with thresholding. It is surprising then that comparisons with naive LASSO or other regularizers have been ommitted altogether (in Table 2), and a (somewhat arbitrary) set of benchmarks have been provided (and that too, only for CIFAR-10). A fair comparison would involve adjusting the regularizer (and not the optimization algorithm, which the authors have done in their benchmarks, with comparisons on Adam/RMSProp and SET). - In continuation with the above remark, a number of popular sparse NN training algorithms have been omitted altogether, e.g., https://homes.cs.washington.edu/~kusupati/pubs/kusupati20.pdf, https://arxiv.org/abs/1611.06694, https://www.cv-foundation.org/openaccess/content_cvpr_2015/papers/Liu_Sparse_Convolutional_Neural_2015_CVPR_paper.pdf to name a few. Moreover, the central argument is to learn efficient architectures to replace fully-connected and convolutional networks, yet, there are no comparisons of actual model sparsity and FLOPs, something that is typical in the literature. - Finally, I believe that this research domain has moved on from toy experiments on CIFAR-10 and CIFAR-100, especially for heuristic-based regularization techniques, and the absence of experiments on ImageNet or other learning problems (only image classification is studied) raises questions on the scalability of the proposed algorithm itself.

Correctness: The claims and method seem correct in my appraisal, and the experimental methodology is also typical of the area. This does not imply that the experiments contributions are complete, but that they are sound.

Clarity: The paper is indeed well written and easy to read.

Relation to Prior Work: The topic discussed in the paper is very broad, and concerns a variety of different disciplines and approaches across machine learning. While the authors have done a sparse sampling of some of this research, the experimental comparisons are lacking and feel incomplete.

Reproducibility: Yes

Additional Feedback: ===POST REBUTTAL=== After reading the rebuttal and other reviews, I agree with the authors' point about learning inductive biases, however, I still feel that there needs to be more evidence to truly support that this method does indeed learn transferable inductive biases. Moreover, distinction from LASSO (and why it cannot do the same) is something the authors should definitely explore.


Review 3

Summary and Contributions: The paper proposes using minimum description length (MDL) as the criterion for reducing inductive biases in deep learning. They use convolution operator as an example and try to learn it from the data, by implementing the MDL criterion via beta-LASSO. The method can indeed learn architectures with local connections and improves SOTA for fully-connected NNs on CIFAR-10/100 and SVHN.

Strengths: The paper is addressing an important problem of inducing inductive biases from the data as opposed to expertly inserting them by hand. The authors use a compelling approach to disentangle the effects of connection locality, weight sharing, and depth. The approach, based on the proposed beta-lasso learning rule, is tested using both deep and shallow architectures; extensive empirical investigations of diverse network (hyper)parameters shed insights into their effect on NN performance..

Weaknesses: The title of the paper declares the goal of this research is learning convolutions, i.e. local filters. However, the learnt receptive fields (S-FC with beta-lasso) are only anecdotally illustrated and no deeper analysis is offered. It also not elucidated whether this locality is due to the shift-invariant nature of natural images. Would other sort of data result in local filters? In fact CIFAR and SVHN are not the best datasets to showcase the learning of shift-invariant operators. Moreover, there's no effort to demonstrate that the learnt filters are transferable accros image datasets. Finally The work does not make effort to test whether the conclusions made on small networks and toy datasets are maintained with scaling. Theorem 1 is copied from [27] page 64, but with typos.

Correctness: Yes It's a bit puzzling though why local and convolutional counterparts have different number of parameters. Shouldn't they be same?

Clarity: Mostly yes. In Theorem 1, d(h) is defined as "language" but in equation 1 d(h) seems to be used as the length of the hypothesis. Please, clarify. Lines 178-184 sample an explanation of the prefix-free language form ref[26] but not coherently. Move technicalities to appendix as they confuse more rather than elucidate.

Relation to Prior Work: Yes, but the Theorem 1 and Eq 1 are mostly copied from the textbook [26] and appear to be presented as the authors' original contribution.

Reproducibility: Yes

Additional Feedback: The authors excessively rely on the classification accuracy (on test set) as the criterion for "bridging the gap" between the fully-connected and convolutional nets. Their analysis focuses on model complexity in terms of the number of parameters. However, the double descent phenomenon demonstrates that accuracy is but one cross-section of a more complex picture. Thus, performance depends not only on model complexity, but also data complexity, and training length. While the two points on #epochs (500 and 5000) are given in the table, this is not discussed in the context of the double descent. Line118: based -> base


Review 4

Summary and Contributions: This paper aims at searching convolution networks from the fully connected networks. At first, they used several empirical studies to compare the convolution networks and fully connect networks. With the additional theoretical analysis, the authors concluded that minimum description length is the guiding principle to choose networks. Finally, the author trained the fully connected neural network with a variant of the LASSO algorithm and achieved good performance in CIFAR and SVHN datasets.

Strengths: It achieves state-of-the-art performance with the MLP network on the CIFAR and SVHN dataset.

Weaknesses: The major conclusion, when the model is able to fit the data, the fewer parameters usually generalize better, is not novel to the research community. This idea is widespread since the success of LASSO. Furthermore, a significant advantage of convolution operation is that it can be applied anywhere of targeted architectures and on any image domains. If authors want to prove that the proposed algorithm is able to search a similar operation from the fully connected networks. I recommend adding experiments about stacking the searched operation to a deeper model and proving the searched operation is transferable among different datasets (such as between CIFAR and ImageNet, or between CIFAR and SVHN).

Correctness: Yes

Clarity: Yes

Relation to Prior Work: The authors do a good job of covering the related neural architecture search work. But in my opinion, it may need to cover the related work discussing regularization and generalization in the traditional statistical machine learning. Because Section 3 and 4 have a big overlap with this domain.

Reproducibility: Yes

Additional Feedback: -----after author feedback----------------- I have read the authors' rebuttal and other reviewers' comments. I still have concerns about the transfer ability experiments. So I decide to keep my score.

[Author Response · NeurIPS 2020]

We thank all reviewers for their feedback. We have fixed minor issues/typos raised by reviewers.

**R1 Results on NLP and Audio Tasks** In the limited rebuttal window, we were able to achieve promising results on
MLPs using $\beta$-LASSO on NLP and Audio tasks: Test accuracy on UrbanSound8K (audio classification): a)MLP
($\beta$-LASSO): $61.5\%$ b)MLP (SGD+weight decay): $54.2\%$. Test accuracy on AG News (text classification): a)MLP
($\beta$-LASSO): $91.2\%$ b)MLP (SGD+weight decay): $88.1\%$ (SOTA is XLNet (Yang et. al. 2019) with $95.5\%$). We will
add more results on other domains in the final version. Since this was the only major concern raised by the reviewer, we
respectfully ask them to consider accepting this paper.

**R2 Introducing S-FC, S-CONV** We first want to clarify that as the title suggests, this paper is not about the "impact
of sparsity in training neural networks". It is about understanding the inductive bias of convolutions and moving
towards learning them from data. One of the contributions of the paper is to showcase the success of an algorithm that
encourages sparsity in learning local connectivity. We hope that reviewer would judge the paper according to the main
motivation and contributions of the paper. In section 2.1 we discuss in details why introducing S-FC and S-CONV is
necessary in order to do a *controlled study* on the inductive bias of convolutions.

**Choice of benchmarks** Again as we explain in the introduction, the goal of the paper is to understand the inductive
bias of the convolutions and use that to bridge the gap between MLPs and convnets (which ends up learning local
connections). As a consequence, we compare our method against best reported results on training MLPs. Are are not
aware of any other work that has a competitive result for training MLPs on these datasets.

**Comparison to LASSO and other algorithms/regularizers that encourage sparsity** We have reported the best
known results on training MLPs using *any* algorithm/regularizer. Since our algorithm is a variant of LASSO (which
we use loosely to refer to L1-regularization), we have also compared it in Table 2 ($\beta = 0$ corresponds to LASSO). To
strengthen the results, we will implement a few other algorithms/regularizers that encourage sparsity and test them in
the final version. Thanks for the suggestions!

**This research domain has moved on from toy experiments...** Again we want to emphasize that this paper is not
about presenting yet another algorithm to sparsify networks but it is a study of inductive bias of convolutions and is
presenting sparsity as a way to learn local connectivity from data instead of directly incorporating it in the architecture.
Training MLPs on ImageNet is very challenging and we are not aware of any such successful attempts.

**R3 Theorem 1** MDL theorem is well-known and we only state it formally in Theorem 1 for completion. We have
clearly added ([26, Theorem 7.7]) right before the statement which is the convention to refer to a theorem taken from
another work. As you pointed $d(h)$ in equation 1 should be replaced with $|d(h)|$ to refer to the length of $d(h)$. We will
improve the discussions around Theorem 1 in the final version.

**#param in local and convolutional counterparts** Local and convolution counterparts have the same number of
weights but the convolutional counterpart has less parameters since it uses weights sharing to assign a group of weights
to the same parameter.

**Double Descent** It has been shown that double descent can be mitigated with regularization (Nakkiran et al. 2020). We
do not observe double descent phenomenon in our experiments (not even the epoch-wise double descent). We will add
discussions to address this in the paper.

**R4 It is known to research community that models with fewer parameters generalize better** While this is very
much motivated by VC-dimension and other generalization theory, it is well-documented in deep learning literature that
models with more parameters often generalize better (the opposite of what theory suggests). Here, what we meant is to
show that this deviation from theory only happens in over-parameterized regime and therefore to compare two model
families (based on their generalization), we suggest that one can compare them in under-parameterized regime and the
model family that performs better on under-parametrized regime will also perform better in over-parameterized regime.
We are not aware of any prior work that points to this observation about such scaling behavior of model families in deep
learning. We will add more context to this discussion in the paper to make it more clear.

**Transferability of the search operation** Thanks for these suggestions. We are indeed thinking of these as future
work. However, we want to emphasize on the significance of the current result. Even without going deep, we have
improved the SOTA on training MLPs by 10% on CIFAR10 dataset (to 84.5 using a simple algorithm. Being able to
learn convolutions takes many such papers and we are hoping to inspire the community to work on these directions.
Considering these contributions and with clarifications we have provided, we hope that the reviewer would consider
accepting the paper.

[Meta-Review · NeurIPS 2020]

The reviews for this paper were overall positive. The authors present an algorithm that, without having any bias related to images, allows one to learn locally connected weights, and provide a study of inductive bias of convolutions. The reviewers appreciated the contributions which address an 'important ML problem' and the developments in particular the interesting results obtained with MLPs on CIFAR-10. The reviewers pointed out several venues for improvement, in particular, the positioning of the paper, the empirical evidence provided to support the claims made, and the thoroughness of the empirical evaluation. A reviewer for instance would have liked to see 'experiments about stacking the searched operation to a deeper model and proving the searched operation is transferable among different datasets (such as between CIFAR and ImageNet, or between CIFAR and SVHN' in order 'to prove that the proposed algorithm is able to search a similar operation from the fully connected networks'. A reviewer would have expected comparisons of Beta-LASSO 'with naive LASSO or other regularizers' over a range of values of the regularization. The authors submitted a response to the reviewers' comments, as well as confidential comments to the area chair. After reading the response, updating the reviews, and discussion, the reviewers maintained their overall positive assessment about the paper. They also encourage the authors to follow their suggestions while preparing the final version to make the paper more impactful. Accept.